# Identification and Characterization of a Novel α-L-Fucosidase from *Enterococcus gallinarum* and Its Application for Production of 2′-Fucosyllactose

**DOI:** 10.3390/ijms241411555

**Published:** 2023-07-17

**Authors:** Ziyu Zhang, Yuting Li, Mujunqi Wu, Zhen Gao, Bin Wu, Bingfang He

**Affiliations:** 1College of Biotechnology and Pharmaceutical Engineering, Nanjing Tech University, 30 Puzhunan Road, Nanjing 211816, China; 202161118034@njtech.edu.cn (Z.Z.); 202161118030@njtech.edu.cn (Y.L.); 202061118048@njtech.edu.cn (M.W.); wubin1977@njtech.edu.cn (B.W.); 2School of Pharmaceutical Sciences, Nanjing Tech University, 30 Puzhunan Road, Nanjing 211816, China; bingfanghe@njtech.edu.cn

**Keywords:** α-L-fucosidase, *Enterococcus gallinarum*, transfucosylation, 2′-fucosyllactose

## Abstract

2′-fucosyllactose (2′FL) is an important nutrient in human milk that stimulates beneficial microbiota and prevents infection. α-L-fucosidase is a promising component for 2′FL synthesis. In this study, a soil-oriented α-L-fucosidase-producing strain from *Enterococcus gallinarum* ZS1 was isolated. *Escherichia coli* was employed as a host for cloning and expressing the α-L-fucosidase gene (*entfuc*). The EntFuc was predicted as a member of the GH29 family with a molecular mass of 58 kDa. The optimal pH and temperature for the activity of EntFuc were pH 7.0 and 30 °C, respectively. The enzyme exhibited a strictly specific activity for 4-Nitrophenyl-α-L-fucopyranoside (pNP-Fuc) and had a negligible effect on hydrolyzing 2′FL. EntFuc could catalyze the synthesis of 2′FL via transfucosylation action from pNP-Fuc and lactose. The yield of 2′FL reached 35% under optimal conditions. This study indicated that EntFuc with a high conversion rate is a promising enzyme source for the biosynthesis of 2′FL.

## 1. Introduction

Human milk oligosaccharides (HMOs) are the third largest solid ingredient in human milk and have been proven to be crucial to stimulating healthy microbiota, avoiding infection, and improving immunity [1,2,3]. The presence of HMO is the major difference between breast milk and cow’s milk [4]. The most predominant component of HMOs is 2′-fucosyllactose (2′FL) [α-L-Fuc-(1-2)-β-D-Gal-(1-4)-D-Glc], which accounts for more than 30% of total HMOs [5]. 2′FL was approved as a nutritional ingredient in infant formula, dietary supplements, and medicinal meals by the European Union owing to its numerous health benefits [6]. 

At present, extraction, chemical synthesis, and enzymatic synthesis are used to synthesize 2′FL. However, the extraction of human milk was difficult due to the complex composition of HMOs, and the amount of lactation was changeable. In addition, complicated steps and the use of tedious substances made chemical synthesis inefficient and less environmentally friendly. Compared to extraction and chemical synthesis, enzymatic synthesis is a promising alternative strategy due to its mild reaction conditions, low pollution, short reaction time, and easy purification of products [7,8,9,10,11,12,13]. Consequently, there is growing interest in the development of biotechnology for producing 2′FL.

Glycosyltransferases (EC 2.4) and glycosidases (EC 3.2.1) were used to catalyze the formation of glycosidic bonds to produce 2′FL [14,15]. The capability of fucosyltransferases to synthesize 2′FL is limited due to the difficulties in protein expression, the low soluble component resulting from the enzyme’s membrane-bound nature, and the high cost of sugar nucleotide (GDP-Fuc) [14]. α-L-fucosidases (EC 3.2.1.51) belong to exoglycosidases that facilitate the removal of α-L-fucosyl residue from the non-reducing ends of oligosaccharide chains and synthetic substrates such as 4-Nitrophenyl-α-L-fucopyranoside (*p*NP-Fuc) [15]. In addition, α-L-fucosidases could catalyze the synthesis of 2′FL by transferring fucosyl residue from *p*NP-Fuc to lactose (lac). Based on sequence homology and catalytic structure analysis, the α-L-fucosidases are divided into GH141, GH29, GH151, and GH95. Currently, GH29 and GH95 are the main α-L-fucosidases used to synthesize 2′FL. GH29 α-L-fucosidases consisting of retaining enzymes could cut the glycoside bonds with a double-displacement mechanism in contrast, GH95 α-L-fucosidases are inverting enzymes with a single-displacement mechanism when participating in the reactions [16,17,18]. Owing to this mechanistic distinction, only GH29 α-L-fucosidases with transglycosylation activity are promising for producing 2′FL. 

Several GH29 α-L-fucosidases for synthesizing 2′FL have been reported. For instance, the α-L-fucosidase FgFCO1 from *Fusarium graminearum* was able to produce 2′FL with a conversion rate of 14% [19]. Lezyk et al. isolated α-L-fucosidase Mfuc2 from a soil-derived metagenome library that could catalyze the formation of 2′FL, and the yield was 6.4% from *p*NP-fucose [20]. The GH29 family fucosidase OUC-Jdch16 from *Flavobacterium algicola* exhibited transfucosylation activity to synthesize 2′FL with 84.82% and 92.15% (mol/mol) from *p*NP-fucose within 48 h and 120 h [21]. Although OUC-Jdch16 showed strong synthesis performance, the majority of identified GH29 α-L-fucosidases had low efficiency for synthesizing 2′FL. In addition, although GH29 is the largest fucosidase family, only 34 have been characterized from more than 5000 entries in the CAZY database [18]. Therefore, it is necessary to explore new GH29 α-L-fucosidases with high transfucosylation activity and study the biochemical characterizations of α-L-fucosidases to enhance 2′FL yield.

In this study, a soil-oriented α-L-fucosidase-producing strain from *Enterococcus gallinarum* was isolated, and a novel α-L-fucosidase named EntFuc with high transfucosylation activity was cloned. *Escherichia coli* was used for cloning and expressing EntFuc. The biochemical characterization of EntFuc was investigated. The ability of EntFuc to synthesize 2′FL using lactose and *p*NP-Fuc as substrates was also investigated (Figure 1).

## 2. Results and Discussion

### 2.1. Obtaining α-L-Fucosidase-Producing Strains

In this study, fucoidan was added to enrichment cultures as the sole carbon source to screen the α-L-fucosidase-producing strains. And then samples of the enrichment cultures were spread on an agar plate containing the chromogenic substrate *p*NP-Fuc. Almost 70 strains were grown on the plate, and approximately 56 strains showed obvious yellow reactions, indicating that these strains had high hydrolysis activity of *p*NP-Fuc. Subsequently, the supernatant of lysate and fermentation of these strains were assessed for their capacity to synthesize 2′FL from *p*NP-Fuc and lactose as substrates by the transfucosylation reaction. And among the 56 strains, only eight produced the desired product in the supernatant of lysate. The results indicated that no synthetic activity was detected in the culture supernatants. Without optimizing the reaction conditions, strain ZS1 among the eight strains showed the highest transfucosylation activity to synthesize 2′FL, with a conversion rate of 8% (Table 1). The 16S rDNA genes, which encode the small subunit of rRNA in prokaryotes, have been widely employed to classify organisms and determine evolutionary relationships. To identify the strain ZS1, 16S rDNA sequencing was performed. The 16S rDNA sequence of the strain ZS1 had 98.77% similarity with *Enterococcus gallinarum* (Accession No. NCDO 2313); therefore, this isolated strain was named *Enterococcus gallinarum* ZS1. *Enterococcus* species are recognized as safe microorganisms that are commensal intestinal bacteria in humans and many animals. However, there are very few studies on α-L-fucosidase from *Enterococcus gallinarum* for the synthesis of 2′FL. Therefore, we cloned, expressed, and examined the biochemical characterization of α-L-fucosidase (EntFuc) from *Enterococcus gallinarum* ZS1.

### 2.2. Sequence Analysis of α-L-Fucosidase

By analyzing the annotated genome sequence (Accession No. WP_016612761.1) of *Enterococcus gallinarum*, only a putative α-L-fucosidase was obtained. According to the corresponding α-L-fucosidase gene, the primers were designed to amplify the complete coding sequence of a potential α-L-fucosidase gene from genomic DNA of *Enterococcus gallinarum* ZS1, and then the PCR products were ligated into the pUC18 plasmid, followed by transfer to *E. coli* DH5α cells for sequencing. The amplified gene had an open reading frame (ORF) of 1464 bp encoding 487 amino acid residues with a molecular mass of 58.5 kDa. EntFuc did not contain a signal peptide based on the SignalP analysis, which supported the finding that α-L-fucosidase EntFuc was an intracellular enzyme. The α-L-fucosidase EntFuc was classified as a member of the GH29 family by InterPro. The multiple sequence alignment analysis (Figure 2) showed that the amino acid sequence displayed 98.77% identity with putative α-L-fucosidase from *Enterococcus gallinarum* (Accession No. WP_01661276.1), 98% identity with annotated α-L-fucosidase from *Enterococcus casseliflavus* (Accession No. WP_ 275626525.1), 97.95% identity with putative α-L-fucosidase from *Enterococcus innesii* (Accession No. WP_252709032.1), and 66.8% identity with α-L-fucosidase from *Streptococcus* sp. S784/96/1 (Accession No. WP_162012694.1). Nevertheless, these α-L-fucosidases were putative, and the biochemical properties were not characterized. Among the characterized α-L-fucosidases, EntFuc showed 27.4% identity with the characterized α-L-fucosidase FgFCO1 (Accession No. AFR68935.1) [19]. The homology model of EntFuc was built using SWISS-MODEL. The α-L-fucosidase from *Phocaeicola plebeius* DSM 17135 (PDB ID: 7snk) was used as a template. The architecture of EntFuc is shown in Figure 3a. The putative structure of EntFuc showed the typical GH29 organization, which consisted of an β/α 8-like barrel domain plus a C-terminal β-sandwich domain. According to the sequence alignment of EntFuc with other α-L-fucosidases and the structure comparisons of EntFuc and the template (Figure 3b), two putative active sites belonging to GH29 α-L-fucosidase were identified, including the catalytic nucleophile (Asp236) and the acid/base catalyst (Glu285) [22,23]. The catalytic nucleophile (Asp236) and the acid/base catalyst (Glu285) were located in the substrate-binding pocket.

### 2.3. Expression and Purification of α-L-Fucosidase EntFuc

*E. coli* BL21 (DE3) was used for the expression of the α-L-fucosidase EntFuc with a C-terminal His-Tag from the pET-28a (+) expression vector. The purified EntFuc presented a single band with an apparent molecular weight of 58 kDa on the SDS-PAGE (Figure 4), which is similar to the calculated molecular mass (58.5 kDa). The molecular mass of EntFuc was consistent with the molecular mass of α-L-fucosidases, including Blon_0426 [24] from *Bifodobacterium longum* (50.3 kDa), OUC-Jdch16 [21] from *F. algicola* 12076 (54 kDa), and BT_2970 [17] from *Bacteroides thetaiotamicron* (55.9 kDa), but lighter than the molecular mass of α-L-fucosidase Alf1_Wf [25] from *W. fucanilytica* (67.5 kDa).

### 2.4. Characterizations of α-L-Fucosidase EntFuc for Hydrolytic Activity

*p*NP-Fuc was used as a substrate to determine the hydrolytic activity of EntFuc by measuring the amount of p-nitrophenol at 405 nm using spectroscopy. The maximum hydrolytic activity was found in pH 7.0 phosphate buffer (Figure 5a). The enzyme activity dropped when the pH was below 4.0 and above 7.0. Compared with the highest activity at 7.0, EntFuc retained less than 30% activity in pH 8.0 buffer. The maximum enzymatic activity of GH29 α-L-fucosidase was in the pH range from 6 to 7.5. The α-L-fucosidase EntFuc was stable from pH 5.0 to pH 8.0 (Figure 5b). Previously, the majority of identified α-L-fucosidases displayed the highest hydrolytic activity on *p*NP-Fuc at 37 °C or higher temperatures. The temperature profile showed that the enzyme exhibited the maximum hydrolysis activity at 30 °C (Figure 5c), which was lower than AlfA, AlfB, and AlfC [26] but higher than TfFuc1 [27]. The enzyme EntFuc exhibited the highest activity at relatively low temperatures, which could be because the enzyme originates from *Enterococcus* species. The thermostability of EntFuc was measured from 20 °C to 50 °C for 2 h. About 60% of its original enzyme activity was maintained at 35 °C for 2 h (Figure 5d). The thermotolerance of EntFuc was lower than that of some reported α-L-fucosidases: AfuC [28] from *Streptomyces* sp. 142, Afc2 [29] from *Clostridium perfringens*, and AfcB [30] from *Bifidobacterium bifidum*. Nevertheless, EntFuc showed greater thermostability in comparison to α-L-fucosidases from *Bacteroides thetaiotaomicron* [17] and *Wenyingzhuangia fucanilytica* [25]. 

The specific hydrolytic activity of EntFuc was 10.1 U/mg using *p*NP-Fuc as substrate under optimal conditions, which was lower than that of the α-L-fucosidases Mfuc5 [20] (4180 U/mg), TfFuc1 [19] (761 U/mg), and NixE [19] (152 U/mg). In addition, the specific hydrolytic activity of EntFuc was close to that of α-L-fucosidase OUC-Jdch16 [21] (17.11 U/mg). The OUC-Jdch16 exhibited poor hydrolytic activity but showed significantly higher efficiency to produce 2′FL among the characterized α-L-fucosidases, suggesting that the EntFuc may have a high efficiency to synthesize 2′FL.

To investigate the influence of various metal ions and SDS on the hydrolytic activity of EntFuc, multiple ions and SDS were added to the reaction mixture. The hydrolytic activity of the enzyme towards *p*NP-Fuc was inhibited by Zn^2+^, Cu^2+^, and SDS, but different concentrations of these metal ions had little influence on the enzyme activity (Figure 6). The addition of Ba^2+^ and Fe^3+^ increased the activity. When the concentration of Ba^2+^ reached 10 mM, the activity of EntFuc was enhanced by 36%, while when the concentration of Ba^2+^ was 1 mM, the activity of EntFuc was only enhanced by 14%.

The substrate specificity of EntFuc was assessed based on its hydrolysis capacity against *p*NP-α-L-Fuc. Meanwhile, other chromogenic substrates, including *p*NP-α-D-Gal, *p*NP-β-D-Gal, *p*NP-α-D-Glu, *p*NP-β-D-Glu, and *p*NP-α-D-mannopyranoside, were also applied to determine the hydrolysis activity of EntFuc. The released *p*NP was detected by ultraviolet light at 405 nm. Simultaneously, 2′FL and Xyloglucan (XyG) were also used to assay the hydrolysis activity by releasing Fuc, which was then detected by HPLC. The structures of hydrolysis substrates are shown in Figure 7. The enzyme activity was only observed in hydrolyzing *p*NP-α-L-Fuc, while no activity could be detected using other synthetic substrates, which was different from many characterized α-L-fucosidases. For example, PsaFuc [31] and BT_2192 [17] were able to hydrolyze *p*NP-Fuc and *p*NP-β-D-Gal, PbFuc [16] was able to hydrolyze 2′FL and Mfuc5 [19], FgFCOl [19] could hydrolyze XyG. Meanwhile, most α-L-fucosidases were able to hydrolyze 2′FL. Zeuner et al. reported that NixE, Mfuc5, and FgFCO1 [19] show activity on 2′FL; Lezyk et al. reported that Thma, Mfuc1, Mfuc2, and Mfuc7 [20] also act on 2′FL. The proportion of transfucosylation and hydrolysis activity is important to produce 2′FL. The α-L-fucosidase EntFuc facilitated the synthesis of 2′FL because EntFuc had negligible activity on 2′FL and could conduct a reverse hydrolysis reaction.

The kinetic parameters K_m_, V_max_, and K_cat_ were assayed against *p*NP-Fuc under optimal reaction conditions. The K_m_, V_max_ and K_cat_ of EntFuc against *p*NP-Fuc were 1.178 mM, 1.784 µmol/mg/min and 2.5 s^−1^, respectively. The K_m_ of EntFuc was lower than Alf1_Wf [25] (3.30 mM), BT_2970 [17] (1.5 mM), and AlfC [26] (5.2 mM), but higher than TM0306 [32] (0.034 mM) and Blon_0248 [22] (0.131 mM). The K_cat_ of EntFuc was lower than Blon_0246 [22] (4.481 ± 0.329 s^−1^), TM0306 [32] (5.4 ± 0.2 s^−1^), but higher than BT_2970 [17] (1.3 s^−1^), and Blon_0248 [22] (0.110 ± 0.026 s^−1^).

### 2.5. Transglycosylation Specificity and Identification of the Synthesis of 2′FL

EntFuc catalyzed the synthesis of 2′FL using lactose and *p*NP-Fuc as substrates. Under optimal temperature and pH, donor/acceptor ratio, enzyme concentration, and reaction time were optimized for transglycosylation. When the donor/acceptor was 1:50 (Figure 8a) and the enzyme dosage was 0.5 U/mL (Figure 8b), the conversion of 2′FL reached 35% after 12 h at pH 7.0 at 30 °C. When the reaction time was over 12 h, the yield of 2′FL was not noticeably increased, indicating that the enzyme was deactivated (Figure 8c). As shown in Figure 9a, HPLC peaks with retention times of 7.21 min and 8.23 min were 2′FL and lactose. The MS spectra (Figure 9b) revealed a peak of [M + NH_4_]+ ions at a *m*/*z* value of 506.2170, which is in accordance with the molecular mass of the standard 2′FL (488.44). ^1^H NMR spectrometry for the transfucosylation product also suggested the 2′FL was formed catalyzed by EntFuc using *p*NP-Fuc and lactose as substrates (Figure 9c). To date, most α-L-fucosidases have low efficiency for synthesizing 2′FL. For example, PbFuc [1] from *Pedobacter* sp. CAU209 was reported to produce 2′FL and 3′FL with a yield of 14.5%. The α-L-fucosidase FgFCO1 [19] from *F. graminearun* was able to use XyG as a donor and lactose as an acceptor, and the yield of 2′FL was 14%. Mfuc2 [20] isolated from a soil-derived metagenome library could catalyze the formation of 2′FL, and the yield was 6.4% from *p*NP-fucose. Hence, EntFuc showed excellent potential for the production of 2′FL.

Some GH29 α-L-fucosidases have transfucosylation activity towards different saccharide receptors [21]. We selected glucose, galactose, xylose, fructose, and sucrose as different receptors and *p*NP-Fuc as the fucosyl donor to determine the transglycosylation specificity of EntFuc. Then the transfucosylation products of these various acceptors were detected by ESI-MS. According to MS results (Figure 9d), the transfucosylation product was only detected using fructose as a receptor. Nowadays, much attention is focused on different fucosylated glycosides, particularly fucosylated oligosaccharides, which have a variety of nutritional and beneficial functions, including resisting infection, improving immunity, and supporting the nervous system. The α-L-fucosidase PbFuc was capable of synthesizing various fucosylated glycosides, including fucosylated glucose, fucosylated galactose, fucosylated arabinose, fucosylated rinose, fucosylated xylose, and fucosylated maltose [1]. In addition, Zhou et al. used OUC-Jdch16 to synthesize xylose-fucose, galactose-fucose, glucose-fucose, fructose-fucose, mannitol-fucose, sucrose-fucose, and glycerol-fucose. In the future, the structures and biological activities of these new novel glycosides may be investigated, and these might be beneficial for the health industry [21].

## 3. Materials and Methods

### 3.1. Materials

*E. coli* DH5α and the pUC18 vector were used for gene cloning. *E. coli* BL21 (DE3) was used as a host for α-L-fucosidase gene expression. pET-28a (+) was used as the expression vector. A restriction endonuclease kit was obtained from TaKaRa (Tokyo, Japan). 4-Nitrophenyl-α-L-fucopyranoside (*p*NP-Fuc), 4-Nitrophenyl-α-D-galactopyranoside (*p*NP-α-Gal), 4-Nitrophenyl-β-D-galactopyranoside (*p*NP-β-Gal), 4-Nitrophenyl-α-D-glucosaminide (*p*NP-α-Glu), 4-Nitrophenyl-β-D-glucosaminide (*p*NP-β-Glu) and *p*NP-α-D-mannopyranoside were obtained from Sigma Chemical Company (St. Louis, MO, USA). 2′FL was supplied by Sangon Biotech (Shanghai, China). Lactose, glucose, galactose, xylose, fructose, and sucrose were purchased from Sinopharm Chemical Reagent Co., Ltd. (Shanghai, China). 

### 3.2. Microorganism Screening and Identification

Soil samples were collected from Forest Park in Jiangsu Province and suspended in sterile water at 37 °C for 1 h. The soil supernatant was inoculated into the enrichment culture containing 0.2% fucoidan, 0.3% NaCl, 0.5% NaNO_3,_ 0.01% K_2_HPO_4_, 0.01% CaCl_2_, and 0.05% MgSO_4_ by shaking at 37 °C for 48 h. For primary screening, the samples were spread on the medium and incubated at 37 °C for 24 h. The medium was supplied with 1% tryptone, 0.5% NaCl, 0.02% K_2_HPO_4_, 0.5% yeast extract, 0.5% NaNO_3_, 2% fucoidan, 0.04% MgSO_4_, 2% agar, and 20 μg/mL 5-bromo-4-chloro-3-indolyl β-D-fucopyranoside (X-Fuc). The α-L-fucosidase-producing colonies were screened based on the yellow color reaction. The screened colonies were incubated in the Luria-Bertani medium. After shaking at 37 °C for 24 h, the strain was cultured on the fermentation medium (1% tryptone, 0.5% NaNO_3_, 0.5% NaCl, 0.02% K_2_HPO_4_, 1% yeast extract, 2% fucoidan, 0.04% MgSO_4_). Then the cells were collected and lysed by ultrasonication on ice. The hydrolysis activity was assayed using *p*NP-Fuc as a substrate. The transfucosylation activity was determined by *p*NP-Fuc and lactose at 30 °C for 12 h. After PCR-amplification reactions, the 16S rDNA sequences were submitted to BLAST, and the strain with the highest transfucosylation activity to synthesize 2′FL was selected.

### 3.3. The Sequence and Structure Analysis of α-L-Fucosidase

The ExPASy-Translate server (http://www.expasy.org, accessed on 9 August 2022) was used to translate the DNA sequence into an amino acid sequence. BLAST (https://blast.ncbi.nlm.nih.gov/Blast.cgi, accessed on 9 August 2022) was used to search for protein homology. The potential signal peptide was predicted using SignalP5.0 (https://services.healthtech.dtu.dk/services/SignalP-5.0/, accessed on 9 August 2022). The sequence alignment was conducted using ClustalW (http://www.genome.jp/tools-bin/clustalw, accessed on 9 August 2022), and the parameters were kept as default values. InterPro (http://www.ebi.ac.uk/interp rol, accessed on 9 April 2023) was used to classify EntFuc. From the NCBI CD-Search (https://www.ncbi.nlm.nih.Gov/Structure/cdd/wrpsb.cgi, accessed on 9 April 2023), the conserved domain analysis was studied. The structure model of the α-L-fucosidase EntFuc from *Enterococcus gallinarum* was built by SWISS-MODEL using α-L-fucosidase from *Phocaeicola plebeius* DSM 17135 (PDB-ID:7snk) [33] as a template. The template had 42.70% identity with EntFuc, and the Global Model Quality Estimate (GMQE) of the template was 0.77. The software Autodock was used to perform molecular docking. The *p*NP-Fuc was docked into the binding site of the EntFuc using α-L-fucosidase from *Phocaeicola plebeius* DSM 17135 (PDB-ID:7snk) as a template.

### 3.4. Cloning, Expression, and Purification of α-L-Fucosidase

According to the DNA sequence of a putative α-L-fucosidase from *Enterococcus gallinarum* (Accession No. WP_016612761.1), two specific PCR primers with 5′-CATGCCATGGTTAGCACGCCTTCAGCCTCC-3′ and 5′-ATTCGAGCTCCGTCGACAAGCTTGC-3′ were used to amplify the complete open reading frame of the potential α-L-fucosidase gene using the genome of the target strain as the template. The constructed plasmid pUC18-EntFuc was transferred to *E. coli* DH5α for sequencing.

The α-L-fucosidase gene *entfuc* was ligated into the pET-28a (+) vector and transformed into *E. coli* BL21 (DE3). The recombinant plasmid pET 28a-*entfuc* was cultured in Luria-Bertani medium containing 50 μg/mL kanamycin by shaking at 37 °C prior to induction. Optical density at 600 nm (OD_600_) of the growing culture was measured, and at an OD_600_ of 0.6–0.8, expression of recombinant protein was induced by the addition of isopropyl-β-D-thiogalactopyranoside (IPTG) at a final concentration of 0.1 mM. Cell cultures grew for 12 h at 16 °C. Then cells were lysed by ultrasonication and centrifuged at 12,000 rpm for 15 min at 4 °C. The supernatant as crude enzyme was collected by centrifugation and then purified by sterile filtration through a 0.22 μm filter. To obtain the purified protein, the Ni-NTA agarose column for affinity chromatography was equilibrated with 20 mM Tris-HCl (pH 7.5). Bound protein was eluted with 20 mM Tris-HCl (pH 7.5, 500 mM imidazole) at a rate of 2 mL/min. The purified EntFuc was analyzed by 12% sodium dodecyl sulfate-polyacrylamide gel electrophoresis (SDS-PAGE) [34]. The protein concentration was determined by the BCA kit [35]. 

### 3.5. Enzyme Assay

*p*NP-Fuc (3.5 mM) was mixed with the EntFuc in 50 mM phosphate buffer (pH 7.0) at 30 °C for 10 min (220 μL reaction mixture) and stopped by adding an equal volume of 1 M sodium carbonate [36]. The amount of p-nitrophenol was measured at 405 nm (Shimadzu UV-160A, Tokyo, Japan). The definition of enzyme activity (U) was the amount of enzyme required to produce 1 μM of p-nitrophenol in one minute.

### 3.6. Characterizations of α-L-Fucosidase EntFuc for Hydrolytic Activity

*p*NP-Fuc was used as a substrate to determine the hydrolytic activity of EntFuc by measuring the amount of p-nitrophenol at 405 nm using spectroscopy. The maximum hydrolytic activity was found in pH 7.0 phosphate buffer. The optimal pH of α-L-fucosidase EntFuc towards *p*NP-Fuc was determined using various pH values ranging from 3 to 10 at 30 °C. The reactions were performed with the following buffers: citrate buffer (pH 3–6), phosphate buffer (pH 6–8), and Tris-HCl (pH 8–10). The pH stability of the enzyme was evaluated by incubating it in different buffers for 2 h and residual activity was measured. The hydrolysis activity of EntFuc was measured at different temperatures, from 20 °C, to 60 °C to determine the optimal temperature. The thermostability was determined by measuring the residual activity after pre-treatment at various temperatures for 2 h.

The effects of metal ions and SDS on the hydrolytic activity against *p*NP-Fuc were determined by pre-incubating different metal ions (Zn^2+^, Cu^2+^, Ca^2+^, Mg^2+^, Ba^2+^, Fe^3+^, 1 and 10 mmol/L) and SDS (1 and 10 mmol/L) in 50 mM, pH 7.0 phosphate buffer at 30 °C. Without any ions or SDS added into the reaction mixture as the control reaction. The residual activity was measured by the standard assay and then presented as a percentage relative to the initial activity.

The substrate specificity of hydrolysis activity was assayed using synthetic substrates including *p*NP-α-Fuc, *p*NP-β-D-gal, *p*NP-α-D-gal, *p*NP-β-D-Glu, *p*NP-α-D-Glu, and *p*NP-mannopyranoside (3 mM in 50 mM, pH 7.0 phosphate buffer). Meanwhile, the hydrolytic activities of 2′FL and citrus xyloglucan were also measured by releasing fucose.

The kinetic parameters of EntFuc were measured in a reaction solvent consisting of 1–10 mM *p*NP-Fuc dissolved in 50 mM phosphate buffer (pH 7.0) at 30 °C for 10 min. And stopped by adding an equal volume of 1 M sodium carbonate. Then K_m_ and V_max_ were fitted with the Michaelis-Menten curve by Prism GraphPad software.

### 3.7. Acceptor Specificity of EntFuc in Transfucosylation Reactions

*p*NP-Fuc (20 mM) was used as a fucosyl donor and glucose, galactose, xylose, lactose, fructose, and sucrose as acceptors (1 M) to determine the transglycosylation specificity of EntFuc. The reaction mixtures contained 0.3 U/mL EntFuc (50 mM phosphate buffer, pH 7.0). The reaction was run for 12 h under optimal pH and temperature and stopped by adding an equal volume of 1 M sodium carbonate. After that, the transglycosylation specificity of different acceptors was determined by ESI-MS.

### 3.8. Synthesis of 2′FL and Optimization of 2′FL

*p*NP-Fuc (20 mM), lactose (1 M), and 0.3U/mL enzyme EntFuc (50 mM phosphate buffer, pH 7.0) were added into the reaction mixture to synthesize 2′FL. The donor/acceptor ratio, enzyme concentration, and reaction time were optimized to enhance the conversion rate of 2′FL. In detail, the donor/acceptor ratio, including 1:10, 1:20, 1:30, 1:40, 1:50, 1:75, and 1:100, was determined to assay the conversion ratio. Meanwhile, the enzyme concentrations, including 0.1 U/mL, 0.2 U/mL, 0.3 U/mL, 0.4 U/mL, 0.5 U/mL, 0.75 U/mL, and 1 U/mL, were assayed. The effects of reaction time on the accumulation of 2′FL, including 1 h, 2 h, 4 h, 8 h, 12 h, 18 h, and 24 h, were also measured.

### 3.9. Identification of Transglycosylation Products

The HPLC analysis was performed using Aminex HPX-87H (column, 300 × 7.8 mm^2^, Bio-Rad Ltd., Hercules, CA, USA) maintained at 50 °C, and transglycosylation products, substrates, and the 2′FL standard were detected by refractive index detector (RID). The mobile phase was 5 mM sulfuric acid at a flow rate of 0.5 mL/min [37]. The conversion ratio was calculated according to the internal standard method. The mass spectra (MS) analysis was conducted by Agilent 6530 Q-TOF with positive-ion mode infusion/offline electrospray ionization (ESI) to determine the mass of transglycosylation products. The spectrum of MS ranged from 200 to 600 *m*/*z*, and the fragmentation voltage was 170 V. The transglycosylation products and 2′FL standard were dissolved in D_2_O and then transferred into a nuclear magnetic resonance (NMR) tube. One-dimensional spectra were recorded at 400 MHz at 298K for ^1^H. The ratio of the amount of the 2′FL (mM) and the added amount of the *p*NP-Fuc (mM) was defined as the conversion ratio (%).

## 4. Conclusions

Nowadays, the enzymatic synthesis of 2′-fucosyllactose has drawn a lot of interest owing to gentle reaction conditions and environmental friendliness. The glycosidases play significant roles in the synthesis of 2′-fucosyllactose due to the specific synthesis of glycoside linkages. However, there are some limitations in the production of 2′FL using α-L-fucosidase. As is known, α-L-fucosidases have natural hydrolysis activity towards the product in the transfucosylation reactions. It is clear that decreasing hydrolysis activity might have an impact on the final yield of 2′FL. To date, the reported α-L-fucosidases that were able to synthesize 2′FL showed low efficiency owing to the high hydrolysis activity towards products. Therefore, it is necessary to explore a novel α-L-fucosidase with high transfucosylation activity to synthesize 2′FL.

Here, a novel GH29 α-L-fucosidase named EntFuc from *Enterococcus gallinarum* has been identified and characterized. EntFuc originated from *Enterococcus* species, which are typical lactic acid bacteria and important in food and clinical microbiology. Characterizations of hydrolytic activity (optimal temperature, pH, temperature stability, pH stability, metal ions, and chemicals) of EntFuc were performed. In order to find the best conditions for transfucosylation reactions, the donor/acceptor ratio, enzyme concentration, and reaction time were optimized. In addition, the EntFuc exhibited negligible activity to hydrolyze 2′FL and showed high transfucosylation activity to *p*NP-Fuc and lactose. The conversion ratio reached 35% within 12 h under optimal conditions. This study of the biochemical characterizations of α-L-fucosidases EntFuc and exploration of the potential ability to produce 2′-fucosyllactose will offer assistance in synthesizing the valuable and functional fucosylated oligosaccharide. In addition, the origination of EntFuc will provide new resource to exploit new α-L-fucosidases.

## Figures and Tables

**Figure 1 ijms-24-11555-f001:**
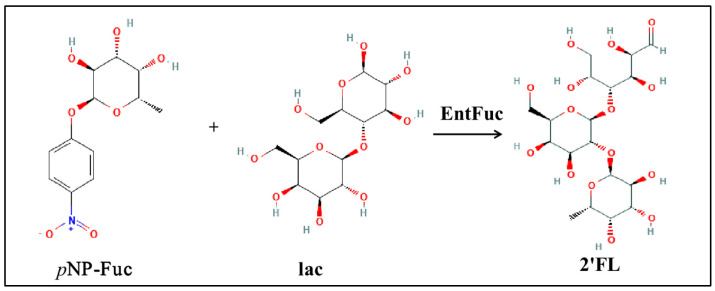
The α-L-fucosidase EntFuc catalyzed the synthesis of 2′−fucosyllactose (2′FL) using 4−Nitrophenyl−α−L−fucopyranoside (*p*NP−Fuc) and lactose (lac) as substrates.

**Figure 2 ijms-24-11555-f002:**
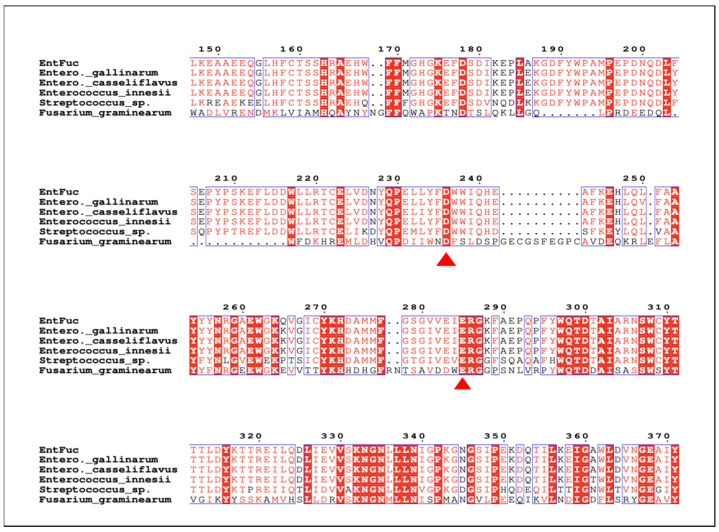
Multiple sequence alignment (MSA) of EntFuc and other α-L-fucssidases. Proteins in the alignments were GH29 α-L-fucosidases from *Enterococcus gallinarum* (Accession No. WP_01661276.1), *Enterococcus casseliflavus* (Accession No. WP_275626525.1), *Enterococcus innesii* (Accession No. WP_252709032.1), and FgFCO1 from *Fusarium graminearum* (Accession No. AFR68935.1). The putative nucleophile residue and acid/base catalytic residue were labeled with red triangles.

**Figure 3 ijms-24-11555-f003:**
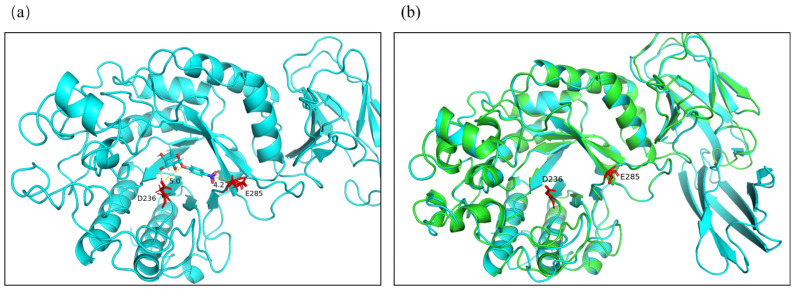
(**a**) Homology modeling of EntFuc using α-L-fucosidase from *Phocaeicola plebeius* DSM 17135 (PDB ID:7snk) as a template. The template had 42.70% identity with EntFuc, and the Global Model Quality Estimate (GMQE) of the template was 0.77. The crystal structure of EntFuc with *p*NP-Fuc docked into the binding pocket are shown. The amino acids Asp236 and Glu285 representing catalytic nucleophiles and acid/base catalysts are shown as sticks, respectively. The distances of the catalytic nucleophile and acid/base catalyst with *p*NP-Fuc were labeled. (**b**) The structure comparisons of α-L-fucosidase EntFuc and template.

**Figure 4 ijms-24-11555-f004:**
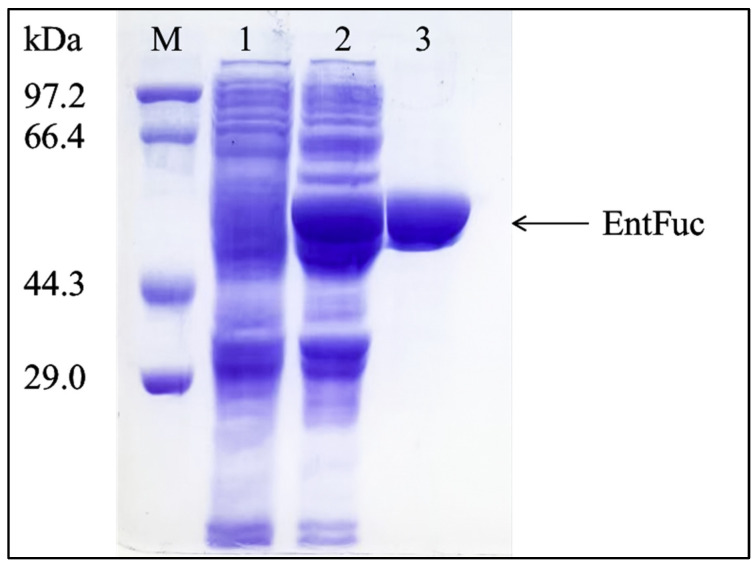
SDS-PAGE of the purified EntFuc. Lane M: standard marker; Lane 1: pET-28a (+); Lane 2: crude α-L-fucosidase EntFuc; Lane 3: EntFuc purified by Ni-affinity chromatography.

**Figure 5 ijms-24-11555-f005:**
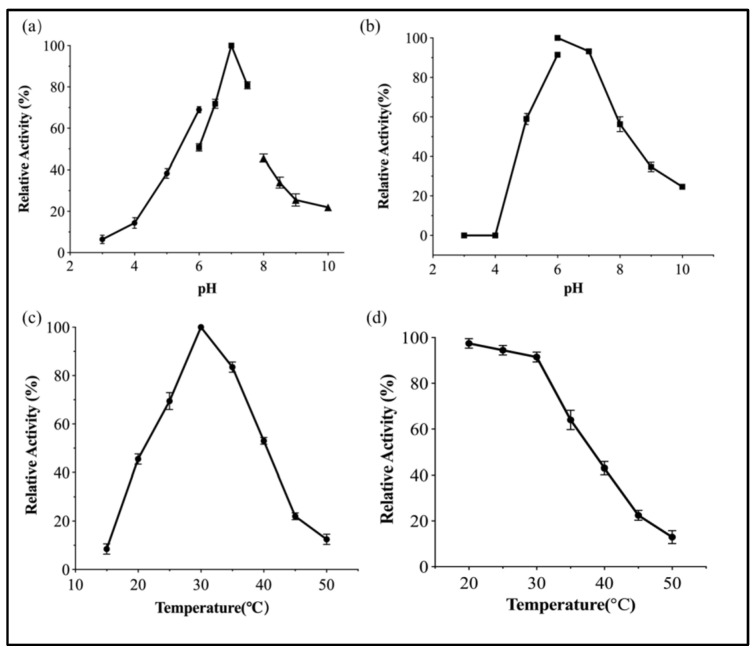
(**a**) The effects of pH on the activity of EntFuc.
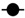
, 
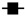
, 
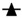
 represent pH 3–6, pH 6–8, and pH 8–10, respectively. (**b**)The pH stability assay. (**c**) The effects of temperature on the activity of EntFuc. (**d**) The temperature stability assay.

**Figure 6 ijms-24-11555-f006:**
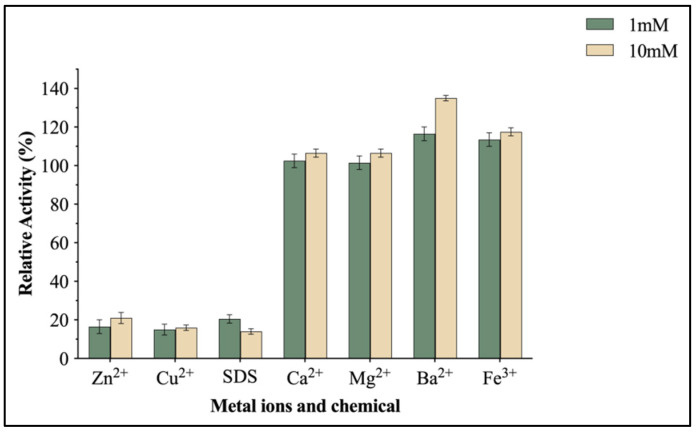
Effects of metal ions (Zn^2+^, Cu^2+^, Ca^2+^, Mg^2+^, Ba^2+^, Fe^3+^) and chemicals (SDS) on the hydrolysis activity of EntFuc. The residual activity was measured by the standard assay and then presented as a percentage relative to the initial activity.

**Figure 7 ijms-24-11555-f007:**
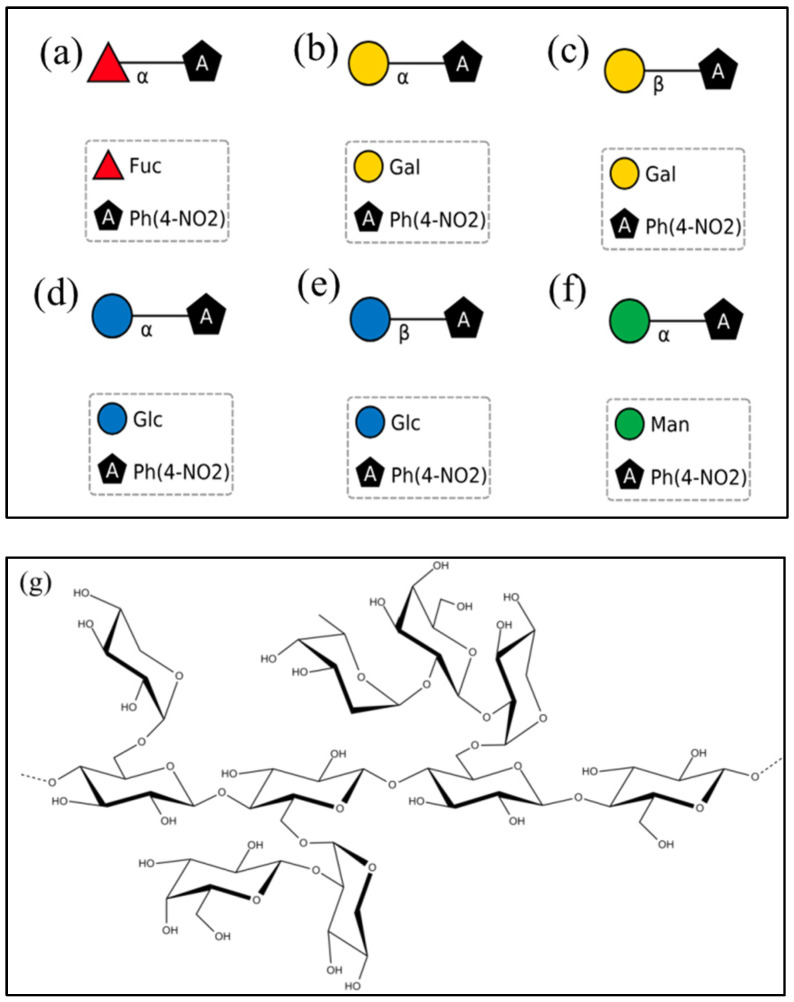
The structure of hydrolysis substrates. (**a**) *p*NP-α-L-Fuc, (**b**) *p*NP-α-D-Gal, (**c**) *p*NP-β-D-Gal, (**d**) *p*NP-α-D-Glu, (**e**) *p*NP-β-D-Glu, (**f**) *p*NP-α-D-mannopyranoside, (**g**) Xyloglucan.

**Figure 8 ijms-24-11555-f008:**
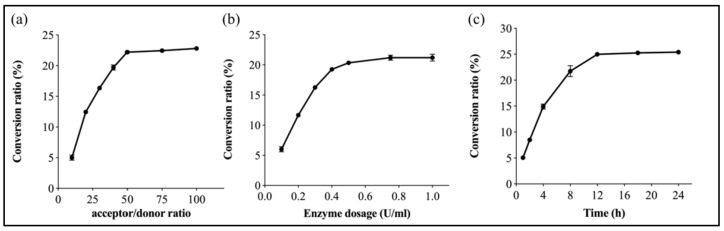
Effects of (**a**) acceptor/donor ratio, (**b**) enzyme dosage, and (**c**) reaction time on the synthesis of 2′FL. Donor/acceptor ratios examined included 1:10, 1:20, 1:30, 1:40, 1:50, 1:75, and 1:100. *p*NP-Fuc (20 mM) and 0.3 U/mL EntFuc were added to reaction mixture. The reaction time was 12 h. The influence of enzyme dosage (U/mL) ranges from 0.1 U/mL to 1 U/mL. 20 mM *p*NP-Fuc and 1 M lactose were added to the reaction mixture, and the reaction was performed for 12 h. The effects of reaction times including 1 h, 2 h, 4 h, 8 h, 12 h, 18 h, and 24 h were assayed on the synthesis of 2′FL. 20 mM *p*NP-Fuc, 1 M lactose, and 0.3 U/mL EntFuc were added to the reaction mixture. The ratio of the amount of the 2′FL (mM) and the added *p*NP-Fuc (mM) was defined as the conversion ratio (%).

**Figure 9 ijms-24-11555-f009:**
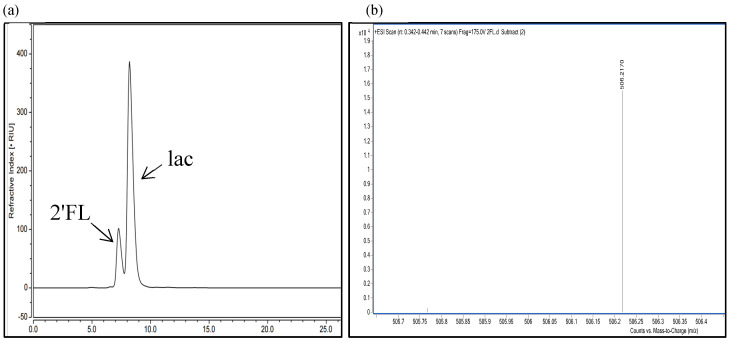
(**a**) The substrate lactose and transfucosylation product 2′FL were detected using HPLC with a refractive index detector. (**b**) MS spectra of the transfucosylation product catalyzed by EntFuc using lactose and *p*NP−Fuc as substrates. (**c**) One−dimensional ^1^H NMR spectra of transfucosylation product. (^1^H ΝΜR (400 ΜHz, Deuterium Oxide) δ 5.25 (d, *J* = 2.6 Hz, 2H, H-1, 7), 5.16 (d, *J* = 3.8 Hz, 1H, H-13), 3.85–3.78 (m, 4H, H-3, 5, 9, 14), 3.76–3.71 (m, 7H, H-2, 8, 10, 11, 15, 16, 17), 3.69–3.58 (m, 4H, H-6, 12), 3.43–3.39 (m, 1H, H-4), 1.16 (d, *J* = 6.6 Hz, 3H, H-18). (**d**) MS spectrum of the transfucosylation product using fructose as acceptor.

**Table 1 ijms-24-11555-t001:** The conversion rate of eight strains with transfucosylation activity. The conversion rate was defined as the ratio of the amount of the 2′FL (mM) and the added amount of the *p*NP-Fuc (mM).

Strains	Conversion Rate (%)
ZS0	3.4
ZS1	8.0
ZS2	3.6
ZS3	4.3
ZS4	5.2
ZS5	2.7
ZS6	<2.0
ZS7	<2.0

## Data Availability

All data included in this study are available from the corresponding author by request.

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
