# Peer review of "Identification and Characterization of a Novel α-L-Fucosidase from Enterococcus gallinarum and Its Application for Production of 2′-Fucosyllactose"

_ijms, 2023, doi:10.3390/ijms241411555_

Round 1

Reviewer 1 Report

In the manuscript submitted as research article by Ziyu Zhang et al., the authors describe the identification, recombinant expression and characterization of a novel α-L-fucosidase belonging to family GH29. The authors further studied the transglycosylation activity of this enzyme aiming to produce human milk oligosaccharides. In general, the outline of the manuscript is scientifically sound although I think some additional experimental details and data need to be provided to make it more comprehensive. In my opinion, the article in its current form suffers from multiple weaknesses. Several points need to be improved before the manuscript meets the quality criteria for publication. Furthermore, I have some specific concerns that need to be addressed before I could recommend the manuscript for publication.

 I hope that the following critiques will aid the authors in refining and revising their manuscript:

Major:

·         The authors describe the isolation of strains from soil samples and they identified 8 fucosidase producers in their initial plate screening. How many different strains/colonies did the authors screen in total? How abundant is fucosidase activity among soil bacteria? Next, the authors selected only the strain with the best transfucosylation activity. How was the transfucosylation of the other strains that showed positive fucosidase activity. I think these results need to be outlined AND discussed in more detail.

·         Sequence analysis: Here the similarity to other α-L-fucosidase sequences is described (Line 78f.) Which of the mentioned enzymes is biochemically characterized? Or are the enzyme activities just predicted from sequence similarity? Could the authors comment on that?

·         Homology modeling: The authors used SwissModel for the prediction of the structure of EntFuc. In my opinion, the authors need to describe the quality of the obtained model in more detail (GMQE, sequence identity with structure template, etc.). Furthermore, the positions of the residues Asp 236 and Glu 291 that are suggested as catalytic side chains appear to be very far apart in the structural model. In my opinion, this does not support the conclusion of the authors that those residues are the catalytic nucleophile and acid/base catalyst. Could the authors please comment on that. Are the wrong residues depicted in the figure? I would find it also very helpful the show the structure of the enzyme used as template for better comparison.

·         Presentation of data: The presentation of data should be in some cases improved to meet the criteria for publication:

o   In general, the resolution seems to be very low. (Maybe this is a problem with the version I received.)

o   Figure 4a: the different buffers used should be indicated in the figure legend or use different symbols, line styles, etc to show the different buffers in the pH profile.

o   Figure 4b: Why is the axis scaled to -20% activity?

o   Figure 6: I can not read the values on the axis in the MS spectra.

o   I would appreciate presenting Km and vmax values together with kcat and specificity constants. Also, a presentation of these data together with the values from literature (Line 167 – Line 170) would make it easier for the reader.

·         Line 172: “Temperature, pH, donor/acceptor ratio, enzyme concentration, and reaction time were optimized for transglycosylation.” Where are the results? How was the yield for the other conditions? What other conditions were tested?

Minors:

Figure 3: change “KDa” to “kDa”

Line 20: What do the authors mean with “less by-product”? Is the enzyme producing any other by-products beside 2’FL? Or do the authors mean less residual substrate?

Line 70: “…by synthesizing 2'FL from pNP-Fuc to lactose.” – In my opinion this is misleading and needs to be rephrased

Figure 1: So far, the authors use the sequence identifiers in the Figure but only the species names in the caption. It would be very helpful to indicate the species names (or abbreviations) in the MSA. Otherwise, I would strongly recommend to describe the sequence identifiers together with the species names in the figure caption.

Figure 5: Why did the authors investigate the effect of Ba2+ ions? Is there any physiological or technical relevance of barium known in this context?

Line 157: What does XyG stand for? Xyloglucan?

Line 189: “This could be a beneficial fucosylated glycoside and in the future, fructose-fucose can be applied in health industry.” This is very speculative…

Line 232: There is a mistake in the link. Please check and correct

Line 233: Typo in link. Please correct

Line 232f.: Please check and correct the grammar of the sentence.

Line 246: What buffer was used for the purification/protein elution?

Line 251: correct “1M” to “1 M”

minor language editing is required

Author Response

We would like to resubmit our manuscript with the initial title “Identification and Characterization of a novel α-L-fucosidase from Enterococcus gallinarum and Its Application for Production of 2'-fucosyllactose”. We greatly appreciate your detailed, encouraging and very helpful comments and suggestions, which have helped us to improve the quality of the study significantly. In the revised version, we have carefully modified our original submission according to your guidance (changes marked in yellow). 

Reviewer 2 Report

In this manuscript, the authors demonstrated the identification of a novel α-L-fucosidase from Enterococcus gallinarum for its application in the production of 2'-fucosyllactose, an important nutrient in human milk. They also expressed the enzyme in E. coli and characterized it after isolation and purification.

The topic of the presented study is interesting and worthy of attention. However, the description of their work is not good and therefore I have some important comments:

-      The authors describe the enzymatic activity of α-L-fucosidase (EntFuc), that is, its ability to catalyze hydrolysis and transfucosylation with different substrates, but there is no scheme of these reactions and no structures of the substrates and products. These should be added in the introduction and according to the place where they are mentioned. The structures of the products, especially 2'FL, should also be presented.

-      The Results section lacks descriptions of screening and identification of microorganisms, plasmid construction, and cloning. Some of these explanations can be found in the Mat and Meth section, where only the technical details should be listed, but the entire thought process and actions should be presented in Results.

-      Results should clearly describe why the sequencing of the 16S rDNA of strain ZS1 was performed (line 72).

-      Section 2.4 "Characterization of α-L-fucosidase EntFuc" - it should be carefully described by which method the activity (both hydrolysis and/or transfucosylation) was/were determined (spectroscopy, HPLC or other). It should also be stated which of the two enzymatic activities (hydrolysis or transfucosylation) was determined in Section 2.4 for each parameter studied, including the kinetic parameters - this should be added at the beginning of the section and/or in the description of each activity.

-      The scheme of the structure of the studied substrates used for the hydrolysis reaction (this is the only place where this is mentioned) should be shown (the substrates are listed in line 155) and a description of the reaction that takes place (e.g. what is the living group or the selection in the scheme).

-      In the experiment shown in Figure 5, there is no control reaction. The control reaction should be a reaction (hydrolysis??) without any ions. Please, add this in the proper part in Results. And please indicate what 100% means and how this relative activity was calculated. Also, SDS is not a metal ion, so please change the sentences in lines 188, 154, and 263.

-      The method used to determine the activity of transglycosylation (section 2.5) should be described (was it HPLC?). How were lactose and 2'FL detected (description of the refractive index detector (RID)).? This should be added in the description of the method and in Figure 6a.

-      Only the products are shown in Figure 6a. What are the retention times of the substrates?

-      Please describe more about the MS analysis (type of MS, instrument)

-      The Materials and Methods section lacks a description of the homology modeling method (shown in Figure 2) and the MSA analysis (Figure 1) - programs and parameters used.

-      There is no Discussion section in the article and only a very short Conclusions section with a few sentences. It looks like there is no scientific discussion about the obtained results. It is allowed to combine results and discussion together, so please decide.

Minor comments:

-      Line 31: Please specify the term "extraction" (which extraction, etc...)

-      Please reword the sentence in lines 63-64: "In this study, a soil-oriented GH29 α-L-fucosidase EntFuc with high transfucosylation activity was screened and identified from Enterococcus gallinarum."

-      The full name of pNP-Fuc should be provided in the abstract.

-      Line 95: "The homology model of EntFuc was constructed using α-L-fucosidase (PDB-ID:7snk) as a template" - Please indicate to which organism this homology exists. The same should be indicated in the description of Figure 2.

-      Many abbreviations are not explained, e.g. ORF (line 83), MSA (line 100).

-      Please rephrase the sentence in lines 139-140: "The specific activity of EntFuc was 10.1 U/mg with pNP-Fuc as the substrate under the optimal conditions, it was lower than the α-L-fucosidases Mfuc5 [20] (4180 U/mg), TfFuc1 (761 U/mg), NixE [19] (152 U/mg).”."

-      Line 286: "The rate of 0.5ml/min" - it should read "the flow rate ..."

I found some mistake in English usage:

"The specific activity of EntFuc was 10.1 U/mg with pNP-Fuc as the substrate under the optimal conditions, it was lower than the α-L-fucosidases Mfuc5 [20] (4180 U/mg), TfFuc1 (761 U/mg), NixE [19] (152 U/mg).”

"In this study, a soil-oriented GH29 α-L-fucosidase EntFuc with high transfucosylation activity was screened and identified from Enterococcus gallinarum"

"speed of" instead of "flow rate"

Author Response

(The authors gave the same response as above.)

Reviewer 3 Report

The authors Zhang and co-workers present a study, in which a novel a-fucosidase from a soil-isolated bacterium is described. The identification and expression of a new a-fucosidase is of great interest, as efficiently transglycosylating a-fucosidases are substantially lacking. However, this work has several serious inconsistencies, which need to be sorted out before this paper can be published.

Major revisions:

1. MSA analysis: how were the catalytic residues determined? The nucleophile D236 seems to be well conserved, however, for the putative acid/base E291 this is not the case. Moreover, Figure 2 shows that there is apparently a big distance between these two residues, too long to be certain that this is the catalytic pair of residues. If the catalytic residues are not absolutely clearly identified from the alignment, other analysis should be done, at least in silico (substrate docking to show the interaction of the substrate with the catalytic residues). Also, mutant variants of the enzyme could be prepared to prove the identity of the catalytic residues.

2. For the determination of the pH optima of the enzyme, it is more efficient to use the set of Britton-Robinson buffers, which cover the pH range from 2 to 12.

3. l. 142-144: This is a strange expectation, that the enzymes with lower hydrolytic activity should have higher transglycosylation activity, these activities are not related in the way the authors suggest. Overall, it is really weird that the enzyme could not hydrolyze 2´FL, which is the expected product of the TG reaction. Normally, the yields of the glycosidase-catalyzed transglycosylations are substantially decreased by the concurrent hydrolysis of the substrate and the product.

4. Last but not least, there is no proof that the transglycosylation product is 2´FL. Based on the molecular mass, it could also be 3´FL, these two trisaccharides could not be distinguished by mass spectrometry. Did you test hydrolysis of 3´FL by this enzyme? The structure of the product can be claimed only when it was isolated and determined by NMR, not based on HPLC peaks. Therefore, I strongly recommend the authors to isolate their product and determine its structure appropriately. Moreover, the method of how the yields of the TG reactions were calculated is not described. Was it just based on the area of the HPLC peaks? Then the column must be calibrated to the concentration of the determined compound and an internal standard should be used in the reactions, otherwise the given yields are nonsense numbers.

The English language is primarily appropriate, there are just some minor issues and some sentences should be rephrased:

l. 41: exogylcosyl enzymes - should be exoglycosidases

l. 63: what is soil-oriented fucosidase?

l. 83: "The predicted protein coded 487 amino acid residues..." doesn´t make sense, please rephrase

l. 90-91: "Two common active sites belonging to GH29 fucosidase were identified in the protein sequence".. these should probably be active site residues?

Author Response

(The authors gave the same response as above.)

Round 2

Reviewer 1 Report

The authors have answered all my questions and improved the manuscript. I only have some minor comments that need to be addressed:

Minors:

Figure 7: The authors use the same symbol (shape and color) for glucose (Glc) and mannose (Man). This should be changed.

Figure 8: The experimental details need to be explained in more detail ideally in the figure’s caption. What was the enzyme dosage and time in 8a? What was the time and acceptor/donor ratio in 8b and so on.

Line 253: correct “maybe” to “may be”

Line 302: English style and grammar: Please correct: The sequence alignment was conducted using ClustalW ((http://www.genome.jp/tools-bin/clustalw) and all parameters kept as default values.

Line 328: Correct imidazon to imidazol

Some minor corrections of English grammar and style are required.

Author Response

We would like to resubmit our manuscript with the initial title “Identification and Characterization of a novel α-L-fucosidase from Enterococcus gallinarum and Its Application for Production of 2'-fucosyllactose”. We greatly appreciate your detailed, encouraging and very helpful comments and suggestions, which have helped us to improve the quality of the study significantly. In the revised version, we have carefully modified our submission according to your guidance (changes marked in green).

Reviewer 2 Report

The authors have responded to almost all comments.

However, there are still some deficiencies in their article:

-          Please, specify what 100% means under the description of Figure 6.

-          I found a lot of language errors, e.g. hydrolyzing capacity (should be hydrolysis capacity), hydrolytic substrates (should be hydrolysis substrates of substrates of hydrolysis) and also hydrolytic activity (rather activity of hydrolysis. Find this in the article in many places and change.

-          Line 2017: reword the sentence “The kinetic parameters Km,Vmax and Kcat were assayed against hydrolyzing pNP-Fuc under the optimal reaction conditions.”  

-          Line 328 “imidazon” change to imidazole, I suppose.

-          Any description of flow during any chromatography should be “flow rate” and not the rate of etc… , so find and change them.

-          There is no good term “by using”. It should be by or using or similar. Find these terms in the manuscript and change them.

-          Line 217 and 341: “Hydrolyzing pNP-Fuc” is not correct English and should be changed.

I found many English errors which I listed in my comment the most important . Therefore, all manuscript should be carefully check for English.

Author Response

(The authors gave the same response as above.)

Reviewer 3 Report

The authors have improved the manuscript substantially following the Reviewer´s suggestions; the paper can be accepted for publication.

Minor language issues occur in the text; professional English editing is recommended but not fully necessary.

Author Response

(The authors gave the same response as above.)
